# Monitoring Osseointegration Process Using Vibration Analysis

**DOI:** 10.3390/s22186727

**Published:** 2022-09-06

**Authors:** Shouxun Lu, Benjamin Steven Vien, Matthias Russ, Mark Fitzgerald, Wing Kong Chiu

**Affiliations:** 1Department of Mechanical & Aerospace Engineering, Monash University, Wellington Rd., Clayton, VIC 3800, Australia; 2The Alfred Hospital, 55 Commercial Road, Melbourne, VIC 3004, Australia; 3National Trauma Research Institute, 89 Commercial Road, Melbourne, VIC 3004, Australia

**Keywords:** osseointegration implant, structural health monitoring, vibrational analysis, E-index

## Abstract

Osseointegration implant has attracted significant attention as an alternative treatment for transfemoral amputees. It has been shown to improve patients’ sitting and walking comfort and control of the artificial limb, compared to the conventional socket device. However, the patients treated with osseointegration implants require a long rehabilitation period to establish sufficient femur–implant connection, allowing the full body weight on the prosthesis stem. Hence, a robust assessment method on the osseointegration process is essential to shorten the rehabilitation period and identify the degree of osseointegration prior to the connection of an artificial limb. This paper investigates the capability of a vibration-related index (E-index) on detecting the degree of simulated osseointegration process with three lengths of the residual femur (152, 190 and 228 mm). The adhesive epoxy with a setting time of 5 min was applied at the femur–implant interface to represent the stiffness change during the osseointegration process. The cross-spectrum and colormap of the normalised magnitude demonstrated significant changes during the cure time, showing that application of these plots could improve the accuracy of the currently available diagnostic techniques. Furthermore, the E-index exhibited a clear trend with a noticeable average increase of 53% against the cure time for all three residual length conditions. These findings highlight that the E-index can be employed as a quantitative justification to assess the degree of osseointegration process without selecting and tracing the resonant frequency based on the geometry of the residual femur.

## 1. Introduction

Traditionally, patients with above-knee amputation are treated with the conventional prosthetic socket device, which utilised a fabricated socket as an interface between the residual limb and prosthesis. This method uses a socket to cover the residual limb, distributing the body weight bearing over the entire socket surface through the soft tissue such as muscles and skins [1]. This direct contact and load transfer, between the prosthetic limb and socket, induces bacteria and allergic reactions developing on the skin and nonphysiological pressure on the limb, leading to significant skin problems and residual limb pain [2,3,4]. In the survey by Hagberg and Brånemark [2], 62% and 51% of 90 amputees reported the skin issue and remnant stump pain when standing or walking, respectively.

Transfemoral osseointegration implant (TFOI), which is treated as an alternative to the prosthetic socket system for transfemoral amputees, has demonstrated notable advantages [5]. TFOI treatment includes two major stages. In the first stage, two surgical sessions are implemented. During the first surgery, an implant is inserted into the prepared medullary cavity where the anchored feature, such as a self-tapping screw or slightly curved stem, engages with bone, and then the residual limb is left unloaded for 3 to 6 months [6,7]. During the second surgery, an abutment is inserted into the end of the implant with the distal end of the abutment penetrating out of the skin, allowing the connection of the artificial limb [8,9,10]. In the second stage, patients with TFOI experience a rehabilitation period, which can take up to 18 months, with gradually increased weight bearing and prosthetic activities [7,8,11]. Titanium alloy is the common material for the TFOI due to its excellent biocompatibility for promoting osseointegration, great mechanical properties and high resistance to corrosion [12,13]. Different from the transtissue connection for the sock system, TFOI provides a stable attachment of the remnant stump to the prosthesis with an implant inserted directly into the residual femur [6,14]. Without the interference of the socket, amputees with TFOI have reported improving comfort when sitting and walking, and elimination of skin and pain problems occurring on residual stump [2,8,11]. Moreover, amputees who are fitted with TFOI also demonstrated a significantly greater range of hip motion and improved control of the artificial limb than amputees wearing socket devices [6,7,11,15]. The direct connection between residual femur and prosthesis improved the proprioception of the leg and foot during the movement, helping amputees to regain their ability to attend sports activities [5,8,16].

Even though the TFOI technique has illustrated significant improvements in life quality of amputees over the conventional prosthetic socket, there are several challenges to this novel technique. The major concern is that an extensive period of rehabilitation protocol is required to achieve sufficient osseointegration and to avoid overloading at the femur–implant interface. The restricted loading on the implant during the rehabilitation period may prevent host bone damage and implant loosening. However, amputees with TFOI have reported concern and frustration regarding the slowness of the rehabilitation period [17]. In addition, the time required for the TFOI to achieve adequate strength varied between individuals due to the diversity of host bone quality and patient body weight. Therefore, a reliable assessment of the femur–implant connection is necessary. Various evaluation methods that are currently clinically available, such as magnetic resonance imaging (MRI) and X-ray computed tomography (CT), are known to be subjective since the results are examined and interpreted by an experienced surgeon rather than by quantitative parameters [18,19,20,21]. Furthermore, the accuracy of these methods is limited due to diffraction effects in the presence of metallic implants [20,22,23]. Patients are exposed to the risk of ionising radiation when subjected to radiative imaging [18,23,24]. Therefore, there is a need for a noninvasive and quantitative method to evaluate the degree of osseointegration, thus shortening the rehabilitation period and preventing implant loosening at an early stage or during long-term application.

Vibrational analysis is a nondestructive structural health monitoring method that is widely utilised to assess structural integrity, and it has also been extended to the biomechanical area. This method has been used to evaluate the stability of dental implants [19,22,24,25] and total hip arthroplasty implants [23,26,27,28,29,30]. In in vitro research proposed by Isaacson et al. [24], a strong positive correlation was noted between resonant frequency and push-out force for the dental implant. Research on vibrational methods to assess the degree of osseointegration for transfemoral implants has proven that the change in the dynamic properties of the bone–implant system, such as resonance frequency and vibration modes, could identify the variation of bone–implant interface conditions along with osseointegration progression [10,19,31,32,33]. In vivo research conducted by Shao et al. [10] involved nine tests during the rehabilitation process of a 40-year-old male patient. During the rehabilitation period, resonant frequency gradually increased along with the osseointegration process, except for a reduction at the first weight bearing. Moreover, research by Cairns et al. [19,20] investigated the ability of modal analysis to detect the change at the composite femur–implant interface by varying the implant inserting torque. According to the results, the change in the boundary condition could be detected by the resonant frequency and mode shape over the specific frequency range.

Recently, in an in vitro experiment conducted on the composite femur with a simulated osseointegration process [34], Lu et al. proposed a new vibration parameter energy index (E-index). The result revealed that the E-index was significantly sensitive to the stiffness change at the femur–implant interface regardless of the femur cross-section, which could be utilised as a quantitative justification to aid in detecting the initial stage of osseointegration. This paper investigates the capability of this vibrational analysis method in monitoring the degree of osseointegration with a novel osseointegration implant design. This novel implant design was developed according to a design concept proposed by Russ, Fitzgerald and Chiu (US20200188140) [35,36], which aimed to embed sensors into the implant to assess implant stability under in vivo conditions. In addition to the previous study, this paper mainly focuses on investigating the behaviour of the E-index throughout the progression of the simulated osseointegration characterised by three residual lengths of the femur.

## 2. Materials and Methods

### 2.1. Test Specimens

The investigation reported in this paper focused on work done with three lengths (152, 190 and 228 mm) of residual femur specimens. The geometry of the implant, shown in Figure 1a, was developed based on the amputated femur models with an oval-shape cross-section. This custom-fit implant consists of three components: a cup-shaped extramedullary (EM) strut, an intramedullary (IM) stem and a prosthesis stem. The EM strut provided the initial resistance to the axial and rotational movement, which is an important prerequisite for osseointegration, allowing close apposition of bone to the surface of the IM stem without the interference of soft tissue. After forming a secure connection between femur and implant, the weight-bearing load was applied though the abutment, which connects to the prosthesis stem. However, there are some changes to the implant model that were used in the experiments, as shown in Figure 1b. The implants were 3D-printed using ABS to fit the cross-section of the 3 different lengths of the remaining femur. The stem was extended from the base to provide a loading point for the experiment. The diameter of the IM stem was reduced slightly by 2 mm to provide sufficient space for the application of the epoxy adhesive, simulating the osseointegration process for the in vivo implant [18], as shown in Figure 2. According to the result in [34], the change in the cross-section of the residual femur has limited effect on the accuracy of the vibrational method investigated in this study. Only an oval-shape implant was employed in this experiment to maintain the consistency between each femur specimen. This is because a recent study [34] showed that the primary parameter of interest is the residual length and not the cross-section shape of the implant.

### 2.2. Experimental Setup

A 250 mm long Sawbone^®^ (Philadelphia, PA, USA) composite femur model was fastened rigidly by a vice through the 3D printed adapter shown in Figure 3. The femur mode was clamped at different sections to simulate three length conditions of residual femur, representing the osteotomy levels of 152, 190 and 228 mm, measured from the greater trochanter. The femur–implant system was stimulated by an input loading through the strike point with an instrumented impact hammer (B&K Type 8206). Two unidirectional accelerometers (B&K Type 4507), which were attached to the bottom of the implant at locations S1 and S2, were arranged to measure the acceleration along the y-axis, as illustrated in Figure 3. The accelerometers were intentionally moved to the edge of implant to avoid the contact with extended base. The signal from the accelerometers were captured and analysed with a two-channel B&K Photon + dynamic signal analyser. The FFT analyser was set up with a frequency bandwidth of 14.4 kHz at a frequency resolution of 1.125 Hz and sampling rate of 32,768 samples per second. The spectra achieved a steady pattern after 8 samples. Therefore, each spectrum was averaged over 10 samples to achieve a good signal-to-noise ratio.

A two-part adhesive epoxy with setting time of 5 min (fully cured after 16 h) was used to simulate the osseointegration process [21,34,37,38,39,40,41]. Even though the curing of adhesive epoxy could not replicate the change at the femur–implant interface during the bone apposition, the stiffness change at interface induced by the chemical reaction could represent similar behaviour during the in vivo osseointegration. The two-part adhesive was prepared and applied to the interface between the cortical shell and IM stem because osseointegration occurs mostly on the IM stem surface. The time was denoted as t *=* 0 when the first set of experimental data were recorded.

The strength of the epoxy adhesive employed in the experiments dramatically increased during the first 5 min after mixing, which provided initial bonding at the femur–implant interface. Then, the strength gradually increased, reaching its maximum after 16 h. The experiments mainly focus on the interface stiffness change during the adhesive’s setting time. Hence, the experiments were conducted at 30 s intervals for the first 300 s (i.e., 5 min), thereby giving us 10 sampling points during the initial curing process. The sampling time was then increased to 60 s intervals for the next 840 s into the experiment. Twenty sets of data with 1140 s of cure process were recorded. Each residual length condition was tested 3 times. The results of the experiments were marked as ‘residual length-test number’; for example, 152 mm-2 represented test 2 with a residual length of 152 mm.

The two-sensor arrangement, which measures the dependent variables that represent the dynamic response of the femur–implant system, is consistent with our previous work [34] and is also employed in other research [37,38,42]. Moreover, this method can evaluate the accuracy of the recorded signals and determine the acceptable frequency range of the signals via a coherence function. The coherence function calculated from the two-sensor setup follows:(1)$$\mathrm{Coherece}=\frac{{\left|\bar{G_{11}\left(f\right)}G_{22}\left(f\right)\right|}^{2}}{G_{11}\left(f\right)G_{22}\left(f\right)}$$
where $G_{11}\left(f\right)$ and $G_{22}\left(f\right)$ are the autospectra of sensors 1 and 2, respectively, and $\bar{G_{11}\left(f\right)}$ is the complex conjugate of $G_{11}\left(f\right)$.

By employing the cross-spectrum of the magnitude, which was normalised to the integration of magnitude over frequency range, the particular vibration modes that were sensitive to the stiffness change induced by the adhesive could be identified with a quality factor, and the variation in the magnitude could be examined. The quality factor was defined as follows:(2)$$\mathrm{Quality}\;\mathrm{factor}=\frac{f_{b}}{f_{c}-f_{a}}$$
where $f_{b}$ is the frequency of the resonant peak, $f_{a}$ and $f_{c}$ are the frequency value 3 dB down from the peak value, lower and higher than $f_{b}$, respectively.

In addition to the cross-spectrum, which describes the dependent variables against frequency, the normalised magnitude was also plotted on a logarithmic scale as a function of time in the form of a colormap. The colormap revealed the variation of magnitude along with the simulated osseointegration process at certain time points. In addition, as mentioned in the previous work, an energy-based index, extended from the research by Ong et al. [43,44], Vien et al. [21,45] and Wing et al. [37,42], was used to characterise the vibrational response to the degree of osseointegration. The E-index is defined as the ratio of integration of the normalised magnitude plot from a certain frequency, ranging from the lower frequency bound $f_{0}$ to the target frequency$\;f_{i}$, relative to the whole frequency range ($f_{0}$ to $f_{1}$), enumerated as follows [34]:(3)$$E\left(t\right)=E_{fi}\left(t\right)/E_{total}\left(t\right)$$
(4)$$E_{f_{i}}\left(t\right)=\int_{f_{0}}^{f_{i}}M^{2}\left(f,t\right)df$$
(5)$$E_{total}\left(t\right)=\int_{f_{0}}^{f_{1}}M^{2}\left(f,t\right)df$$
where M(f,t) is the normalised magnitude at frequency f and cure time t, $E_{total}\left(t\right)$ is the integration of normalised magnitude M(f,t) from $f_{0}$ to $f_{1}$ at cure time t and $E_{f_{i}}$ is the integration of normalised magnitude M(f,t) from $f_{0}$ to $f_{i}\;$at cure time t.

The selection of $f_{0}$, $f_{i}$ and $f_{1}$ varies among each residual length condition and is discussed in the Results section.

## 3. Results

### 3.1. Validity of Data Acquired

As discussed in the previous section, the coherence function generated from signal S1 and S2 was used to statistically analyse the validity of the data acquired. The coherence, which was plotted against the adhesive cure time in frequency bandwidth of 14.4 kHz for three residual length conditions, is shown in Figure 4, Figure 5 and Figure 6. For all three conditions, the plots illustrate that the coherence was distributed across a wide frequency range without significant peaks. After 300 s, marked in the yellow dashed line, the resonance peaks appeared at certain frequencies. This result indicates that the resonance modes were suppressed by the damping effect of the adhesive prior to sufficient strength at the femur–implant interface. Moreover, the coherence from 0 to 8000 Hz was generally above 0.8, indicating the validity of the data collected. In addition, the resonance modes located above 8000 Hz are not clearly shown in the plot. For all three conditions, the plots illustrate that the coherence from 0 to 8000 Hz was generally above 0.8, thereby showing the validity of the data collected. Therefore, the upper bond ($f_{1}$) in Equation (5) for cross-spectrum, colormap and E-index were set to 8000 Hz to achieve accurate results. The lower bound for the frequency range is determined by the behaviour of the cross-spectrum of normalised magnitude over cure time.

### 3.2. Cross-Spectrum Analysis

Figure 7 and Figure 8 exhibit the cross-spectrum of the normalised magnitude, which are plotted in the frequency bandwidth of 8000 Hz for the three residual length conditions. The cross-spectrums at cure times 0, 150, 300, 600, and 1140 s aim to show the magnitude change relative to the simulated osseointegration process.

#### 3.2.1. Residual Femur Length of 152 mm

According to Figure 7, at the early stage of the adhesive curing process (before 300 s), the response was flat except for a number of resonant peaks as marked with a vertical dash–dotted line. There are several peaks that can be visually recognised in the cross-spectrum after the adhesive setting time of 300 s, such as 2371, 2327 and 2407 Hz, indicated in the plot. Along with increasing cure time, the quality factor of the selected frequencies is noted to increase with curing time (see Figure 8). The increase in the Q-factor is noticeable at approximately 300 s, which coincides with the setting time of the adhesive. This finding indicates that the change in the interface condition can be detected by the specific resonant modes by tracking the magnitude and quality factor change during the curing time. However, the resonant peaks for frequency above 4000 Hz remain difficult to distinguish by visual inspection. This finding is explained in the Discussion section.

#### 3.2.2. Residual Femur Length of 190 and 228 mm

Similar trends in the resonant peaks were also noted in the cross-spectrum of 190 and 228 mm, shown in Figure 9a,b. In Figure 9a, before the curing time of the adhesive, the spectrum is flat, and the resonant modes are hard to spot in the plots except for the vibration modes located at the lower frequency range (frequency smaller than 1500 Hz). However, upon reaching the end of setting time (300 s), a significant variation in frequency magnitude around 2700 Hz are observed for all three results of 190 mm. The magnitude for selected frequency noted in Figure 9a, gradually decreases along with the degree of simulated osseointegration. The cross-spectrum of 228 mm demonstrates the similar behaviour of magnitude as the other two residual lengths. The resonance modes that were sensitive to the interface condition can be identified based on the appearance and magnitude change in the resonant peaks. Furthermore, the quality factor for 190 and 228 mm, shown in Figure 10, demonstrate a six-fold increase in the Q-factor during the first 400 s of the experiment. This is consistent with the results for a residual length of 152 mm. This finding shows that tracing the variation in magnitude and Q-factor can qualitatively identify the stiffness change due to the simulated osseointegration process under the basis of using specifically selected resonant frequencies.

### 3.3. Time-Progression of Cross-Spectrum

The cross-spectrum is presented as a colormap to demonstrate its progression as the function of simulated osseointegration time to interpret the correlation between dynamic responses to the degree of osseointegration. For osteotomy levels of 152 and 190 mm, during the early stage of simulated osseointegration process, the damping induced by the soft adhesive epoxy at the femur–implant interface impeded the presenting of high-frequency vibrational modes, leading to a flat cross-spectrum, shown in Figure 7a and Figure 9a, and is evident in Figure 11 and Figure 12. The resonant peaks are noted to have developed at approximately 300 s, which is consistent with the 5 min setting time for the adhesive used in the experiments. After 300 s, multiple resonant peaks were observed in the frequency band approximately 1500 to 8000 Hz for both residual length conditions, especially for the peaks located between 2000 to 3000 Hz. It is noted that the magnitude of the modes under 1500 Hz were less affected by the interface condition. Therefore, the lower frequency bound for the E-index, $f_{0}$, was set to 1500 Hz for 152 and 190 mm conditions (refer to Equations (4) and (5)). The objective is to enhance the ability of the E-index to include only the bandwidth relevant to the simulated osseointegration process. Furthermore, to enhance the sensitivity of the E-index, the target frequency$\;f_{i}$ was set to 2500 and 2900 Hz for the 152 and 190 mm conditions, respectively (refer to Equation (4)), to ensure that the frequency range covered by the E-index has large magnitude changes related to the stiffness changes induced by the simulated osseointegration process.

Similar phenomena were also noted in Figure 13 for the osteotomy level of 228 mm. Before 300 s, which is indicated with a vertical yellow dash–dotted line, the vibration modes were hard to determine. With the secure connection engaged at the femur–implant interface by curing of the adhesive after 300 s, the resonant peaks are easily distinguished by visual inspection, except for the modes that are located below 1500 Hz, which were not affected by the change in the interface condition. In addition, according to the cross-spectrum shown in Figure 9b, a large variation in the resonant peaks during the cure time were identified around 2700 Hz. Therefore, the lower bound $f_{0}$, for the E-index of 228 mm was set to 1500 Hz and target frequency $f_{i}$ was set to 2700 Hz (refer to Equations (4) and (5)).

### 3.4. E-Index as Function of Curing Time

The following plots were generated from the E-index formula based on the specific frequency ranges for each remnant femur length. Figure 14, Figure 15 and Figure 16 show that for all three length conditions, even though there are some fluctuations in the E-index, the plots share a general trend that the E-index dramatically increases during the simulated osseointegration time for the first 300 s, then the gradient drops to 0 and stabilises above 0.7, indicating the implant securely bonded with the femur. A significant gradient change occurred around 300 s, which is consistent with the setting time of the adhesive epoxy. Even though some dispersions can be identified in the plots, the average E-index for three residual lengths, shown in Figure 17, demonstrate that the E-index gradually increases asymptotically to a value above 0.8, indicating that the interface stiffness change incurred by the curing of the adhesive is clearly represented by the variation in the E-index through time. In addition, Table 1 demonstrates the change in the E-index at the end of the experiment relative to the value at t = 0. The E-index of the three remnant femur conditions demonstrates an average increase above 50% with a minimum shift of 45.45%. The results clearly identify the significant increase in the value of the E-index during the curing time.

## 4. Discussion

The appearance of the resonant peaks on the cross-spectrum, after the initial integration between femur and implant, indicated that the stiffness change at the femur–implant interface due to the early stage of osseointegration can be monitored by tracking the magnitude and quality factor of certain vibration modes in the cross-spectrum. Furthermore, colormaps, which show the cross-spectrum of magnitude with the additional axis of a continuous curing time, were employed in this research. The variation of the magnitude on the colormap indicates that the excitation of the high frequency modes was hindered by the damping effect of the soft epoxy during the initial stage of the curing process. Then, the curing of epoxy leads to a stiffness increase at the femur–implant interface. A clear step decrease in the magnitude was observed around 300 s; meanwhile, multiple resonant peaks emerged above 1500 Hz. With the additional axis of time, the colormap provides an overview of all the vibrational modes. The transformation of the resonant peaks associated with the simulated osseointegration process can be visually inspected. In addition, the time when this variation occurred was consistent with the adhesive epoxy’s setting time of 5 min, evidence that the colormap has a significant sensitivity to the stiffness change at the interface during the early stage of the adhesive curing process. Furthermore, colormaps for the three residual lengths shared similar behaviour, which implied that the sensitivity was not affected by the length of the residual femur, proving the capability of using colormap to detect the initial bond of the osseointegration process for three lengths of residual femur, without selecting and tracking the specific frequency for each length.

In addition to the colormap, the recognisable increase in E-index during time and the distinct gradient change after 300 s that is consistent with setting time, show that the E-index is sensitive to the stiffness change at the femur–implant interface, incurred by the solidification of the epoxy. This finding suggests that the E-index could be used as a quantitative approach to aid in the assessment of the implant stability at an early stage of osseointegration. Moreover, the results given in Table 1 show that the amplitude of the E-index increased on average more than 50% with a minimum of 45.45%, which is significantly larger compared to the 3% for resonant frequency analysis [10] and 10–47% difference and modal analysis [19] for all three remnant length conditions. This finding indicates that the accuracy of the E-index is regardless length and cross-section geometry of the residual femur [34]. Furthermore, this result proves the potential of the E-index to become a conventional quantitative justification to evaluate the degree of osseointegration. Moreover, by using an integration method such as E-index to capture all resonant frequencies in certain frequency range, there is no need to estimate their modal parameters. Hence, the difficult in selecting and tracing the resonant frequency based on the geometry of the residual femur is reduced.

Even though the magnitude of several vibration modes changed dramatically and became noticeable as the epoxy cured, shown in the cross-spectrum, the resonant peaks above 4000 Hz are remain difficult to identify. A reason is that the adhesive epoxy used in the experiment requires 16 h to reach its maximum strength. It means that although the epoxy provides the initial bond at the interface, allowing the excitation of several resonant modes, the relatively ‘soft’ behaviour when experiment terminated at 19 min, compared to stiffness at fully cured condition (16 h), suppressed high-frequency modes, leading to a flat cross-spectrum after 4000 Hz. In addition, some fluctuations were identified in the E-index during the cure time. The potential reason might be the two-part adhesive epoxy was not properly mixed, or the volume of the epoxy used was slightly different for each experiment. Another reason might be that the adhesive layer applied on the interface was not consistent and uniform across the experiments, which subsequently affected the performance of the epoxy in providing connectivity between the femur and implant. In addition, the experiment time of 19 min is extensively shorter than the fully curing time of 16 h, meaning that the impact from the manual excitation may damage the interface connection during the experiment, leading to an unstable E-index during the cured time. In spite of these considerations, the clear trend and large shift of the E-index as the epoxy cured indicates the potential of this method to be employed as a universal quantitative way to monitor the degree of osseointegration without the interference of the length and geometry of residual femur. Furthermore, based on the novel implant proposed by [35,36], with the embed sensors on the implant, evaluation of the implant stability on the osseointegration process under in vivo conditions is possible. Nevertheless, further investigation of the E-index on identifying the osseointegration or implant failure to validate the robustness and accuracy of this strategy is planned.

## 5. Conclusions

The work presented shows that the degree of osseointegration, which is simulated by using two-part adhesive epoxy, can be assessed with the dynamic response of the femur–implant system. With the dual-sensor measurement method, the input loading applied by the instrumented hammer is not crucial for the assessment. Cross-spectrum and colormaps of the normalised magnitude during the curing time demonstrate significant changes that were related to the increase in stiffness due to the curing of epoxy, indicating that the application of these plots could advance the accuracy of diagnostic techniques. Furthermore, the accuracy and reliability of the E-index from previous work [34] is further investigated in this paper with three residual femur length conditions. The results indicate a clear trend and significant shift averaging 53% of the E-index during the simulated osseointegration for all three lengths. This finding shows the capability of the E-index as a quantitative approach to monitor the degree of osseointegration, without the burdens of selecting and identifying the specific resonant peaks based on the length of residual femur. Future studies could focus on further validation with the mass damping effect due to the soft tissue under an in vitro condition. This could further enhance the reliability and accuracy of the E-index. Moreover, research on combining the E-index method with the novel implant, which integrates the sensors with the structure to assess the osseointegration under in vivo conditions, is an important next-step for this monitoring strategy.

## Figures and Tables

**Figure 1 sensors-22-06727-f001:**
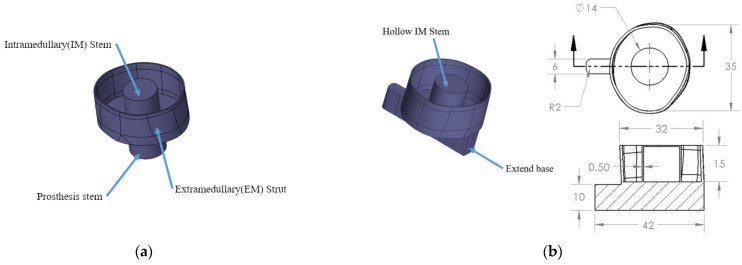
Geometry of the novel osseointegration implant: (**a**) developed based on Patent US20200188140 and (**b**) modified oval-shape implant with hollow IM stem for the experiment, dimension in mm.

**Figure 2 sensors-22-06727-f002:**
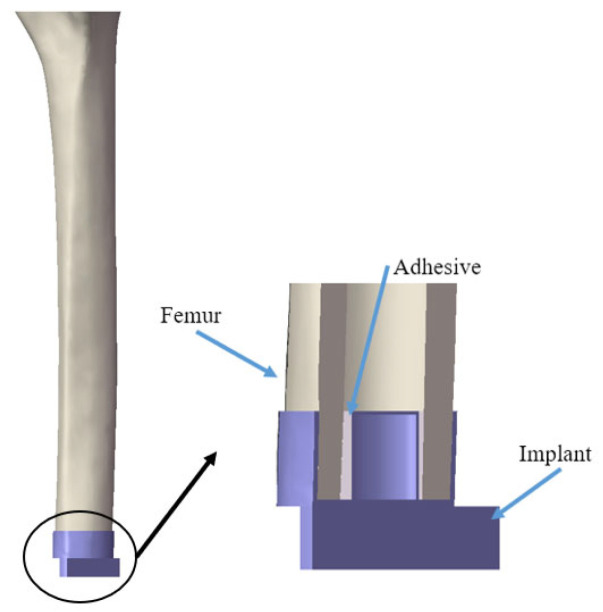
Cross-section of femur–implant interface with adhesive epoxy.

**Figure 3 sensors-22-06727-f003:**
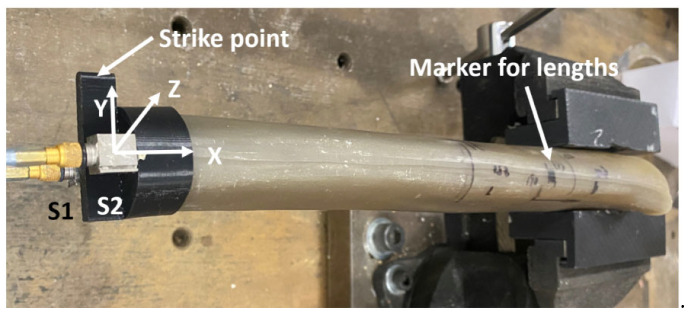
Two-sensor (S1 and S2) setup for composite femur model with markers for three residual length conditions.

**Figure 4 sensors-22-06727-f004:**
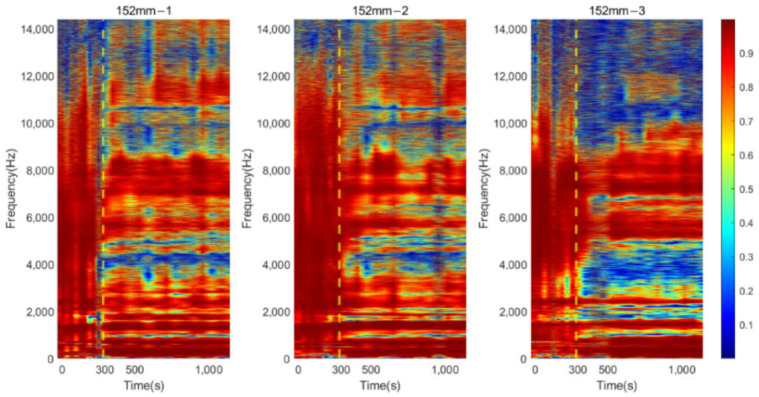
Coherence function for residual length of 152 mm.

**Figure 5 sensors-22-06727-f005:**
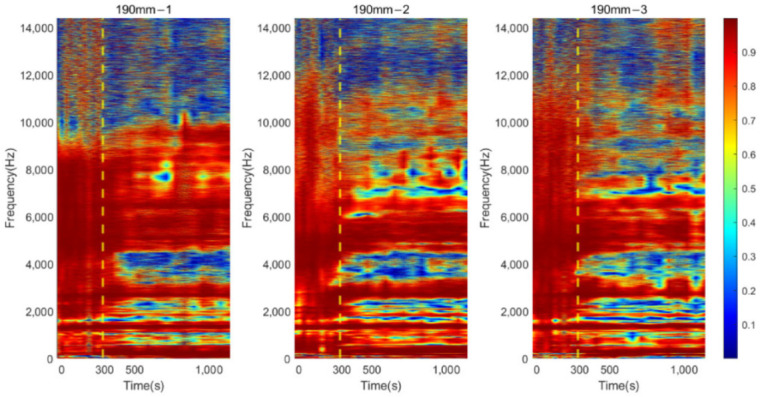
Coherence function for residual length of 190 mm.

**Figure 6 sensors-22-06727-f006:**
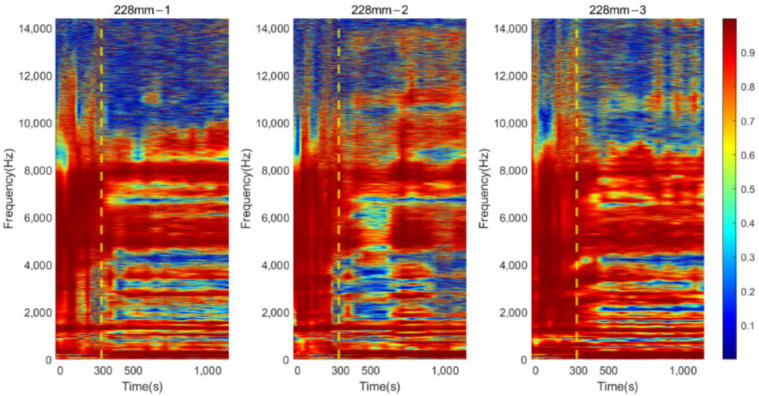
Coherence function for residual length of 228 mm.

**Figure 7 sensors-22-06727-f007:**
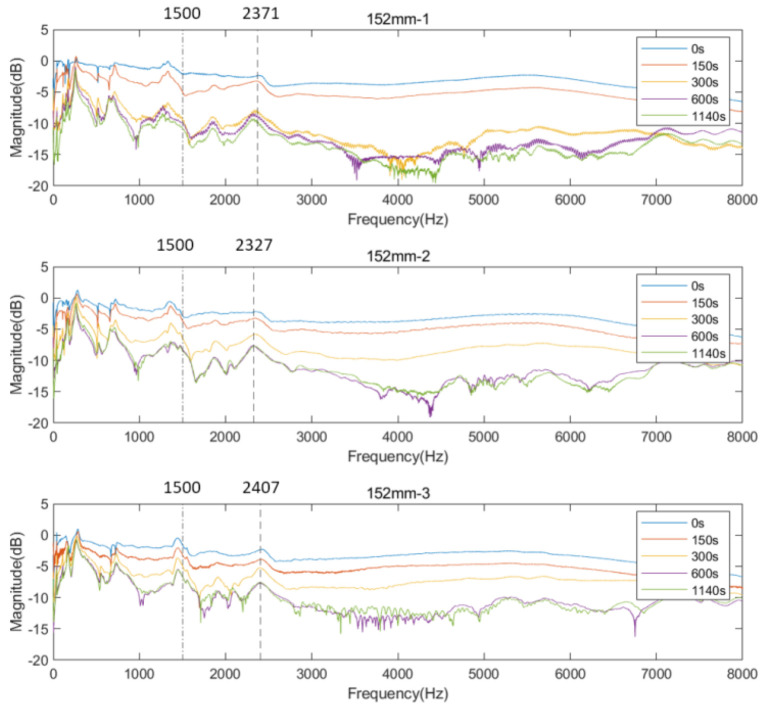
Cross-spectrum of normalised magnitude for the residual length of 152 mm at 0, 150, 300, 600, and 1140 s cure times.

**Figure 8 sensors-22-06727-f008:**
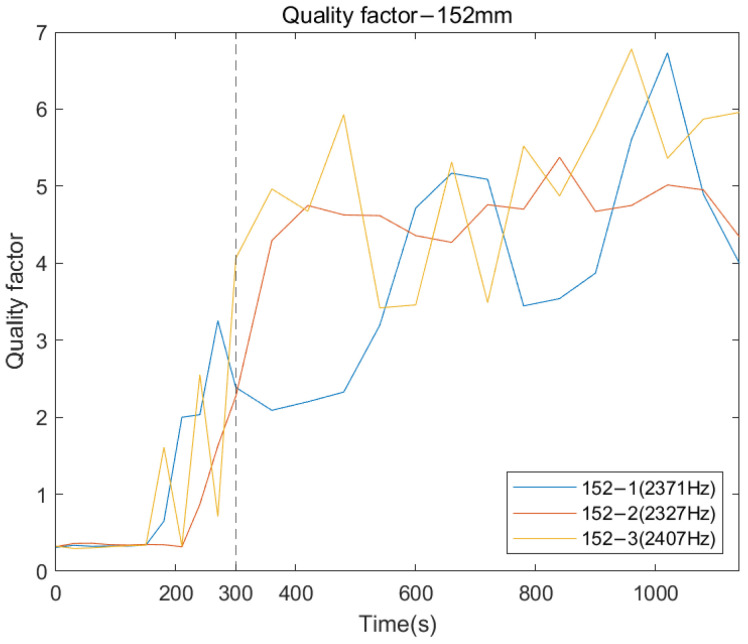
Quality factor of selected resonant peaks as a function of time.

**Figure 9 sensors-22-06727-f009:**
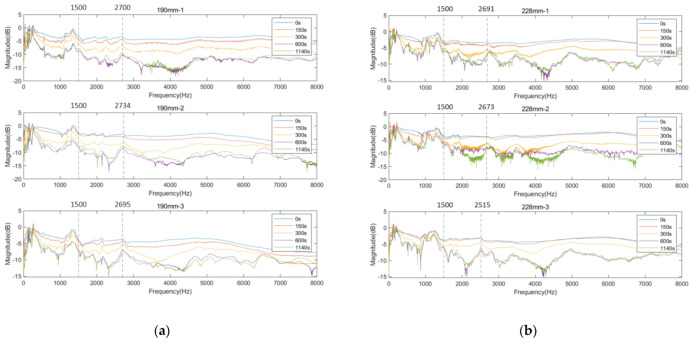
Cross-spectrum of normalised magnitude at 0, 150, 300, 600, and 1140 s cure times for residual lengths of (**a**) 190 and (**b**) 228 mm.

**Figure 10 sensors-22-06727-f010:**
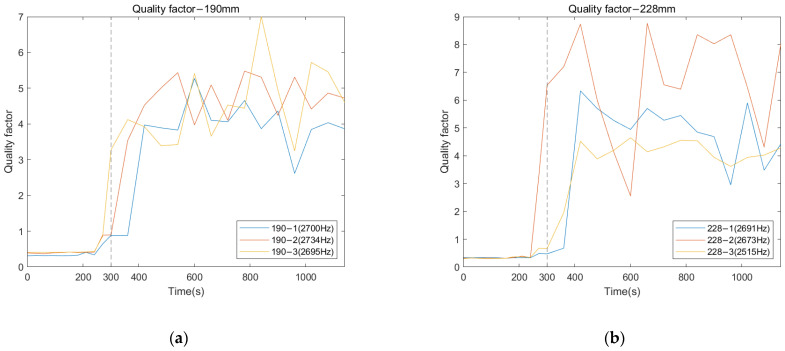
Quality factor of selected resonant peaks as function of time for residual lengths of (**a**) 190 and (**b**) 228 mm.

**Figure 11 sensors-22-06727-f011:**
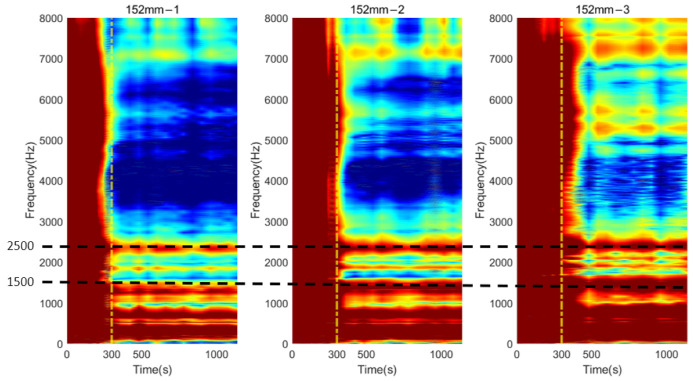
Colormap of the normalised magnitude development as a function of cure time for 152 mm.

**Figure 12 sensors-22-06727-f012:**
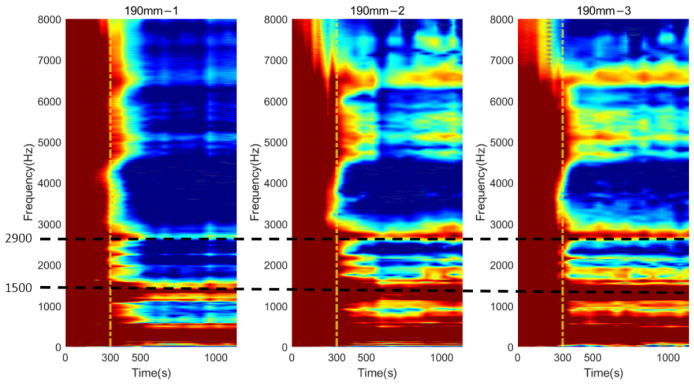
Colormap of the normalised magnitude development as a function of cure time for 190 mm.

**Figure 13 sensors-22-06727-f013:**
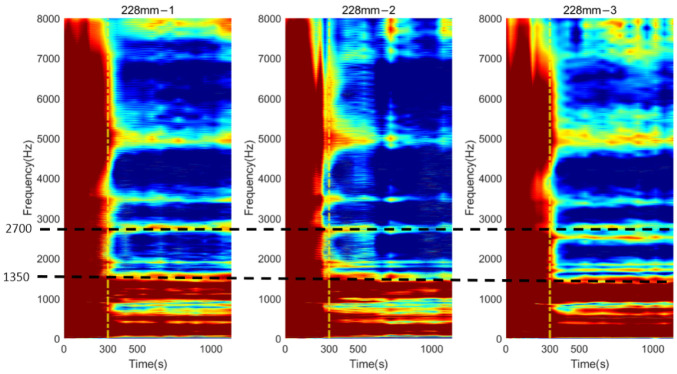
Colormap of the normalised magnitude development as a function of cure time for 228 mm.

**Figure 14 sensors-22-06727-f014:**
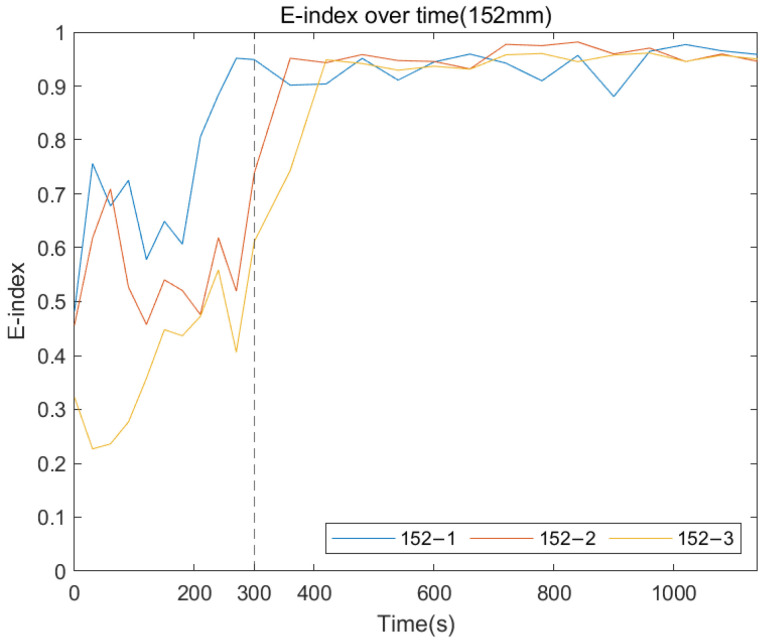
E-index development as the function of cure time for 152 mm.

**Figure 15 sensors-22-06727-f015:**
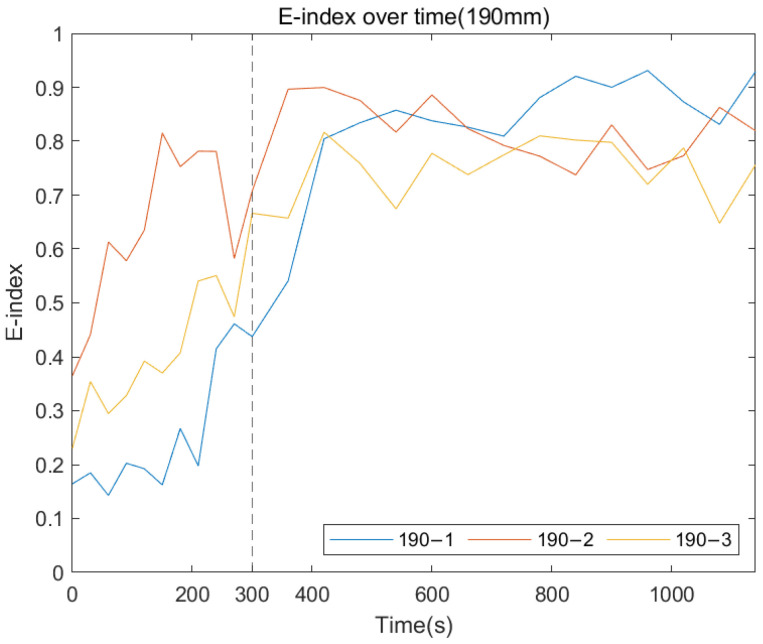
E-index development as the function of cure time for 190 mm.

**Figure 16 sensors-22-06727-f016:**
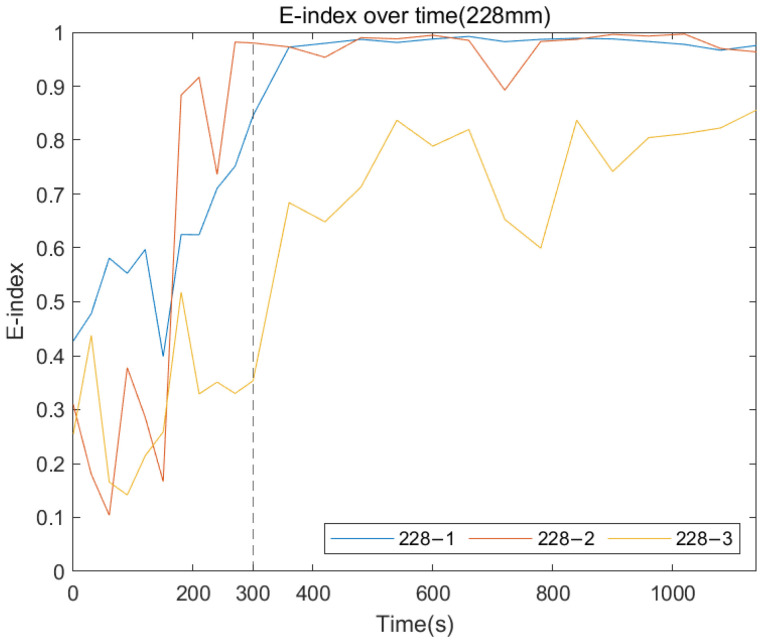
E-index development as the function of cure time for 228 mm.

**Figure 17 sensors-22-06727-f017:**
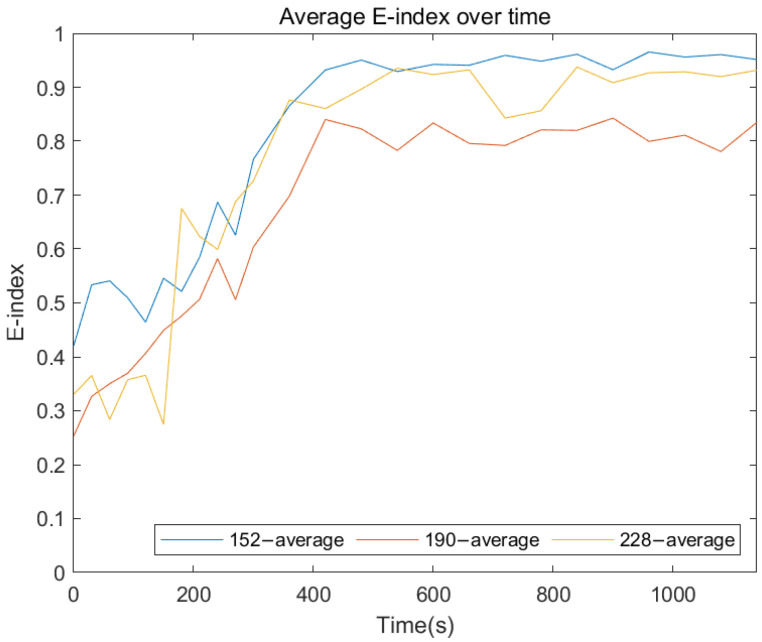
Average E-index development as the function of cure time for three residual lengths.

**Table 1 sensors-22-06727-t001:** Difference of E-index (relative to 0 s) for each length condition.

Test	Difference (%)		
	**152 mm**	**190 mm**	**228 mm**
1	76.49	47.59	54.89
2	45.45	48.89	65.42
3	52.55	62.75	60.56
Averaged	58.16	53.08	60.29

## Data Availability

The raw/processed data required to reproduce these findings cannot be shared at this time as the data also forms part of an ongoing study.

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
