# Peer review of "Monitoring Osseointegration Process Using Vibration Analysis"

_sensors, 2022, doi:10.3390/s22186727_

Round 1

Reviewer 1 Report

In my opinion, authors must modify/clarify the following issues:

Major changes

#1) Fig. 1. Please add units for dimensions shown in the figure and include proper dimensions for the oval shape.

#2) Fig. 2. For the sake of clarity, please include a 3D view of the set up shown

#3) Line 161. Please explain deeply why the times intervals of 30s for the first 300 s and 60 s later are selected. In addition, the limit time of 300 s for measurements and the adhesive setting time of 840s and the final time of 1140 s.

#4) Line 206. For the sake of clarity, please explain why the frequency bandwidth of 14.4 kHz was selected and indicate how the statistically analysis for validating the data were carried out and how the color plots were obtained for measurement data.

#5) A deep discussion of results is needed for figures 4-5 and 6. Why the coherence is reduced with time for higher frequencies (blue color) in some cases and in others not?. This is easy to observe in Figure 6. In addition, Why for a given frequency, for instance, 4000 Hz in fig. 4 the coherence is reduced with time and in other frequency the coherence is similar (5000Hz)?. For the sake of clarity please explain the differences between the three tests since color maps are quite different in certain zones for instance the upper right zone of test 3 of figure 4 is blue whereas the same zone the blue zones are smaller. Please do the same for fig. 5 and 6.

#6) line 228. Why these resonant peaks are selected?. Please explain deeply the oscillating behavior appearing in curves for 4000-5000Hz (test1) and 3500-5000Hz (test3) for the case 152-1 for long times. Please do the same for the other lengths (Fig. 9) especially for the case 228mm-2.

#7) Fig. 9. There is a high dispersion in the 3 tests for each case of study with different tends for each plot, for instance a oscillating curve appears for test 228mm-2 and in the other two it does not appear. Please explain all this deeply.

#8) Line 253. Authors state “dramatical increase in gradient around 300 second ” Please be more precise gradients are commonly used for variations of a variable in one direction, please include the variable with the term gradient. In addition, please use “300 s” instead of “300 second”

#9) Line 254. Authors state “this finding showed that tracing the resonant modes can identify the stiffness increase……” Please explain deeply how this can be done. How resonant mode is traced and how can quantify the stiffness increment. This is a key point.

#10) Figs. 14-15 and 16. In this figures a high dispersion in results can be observed comparing the three test results. Please explain this deeply and, in my opinion, an average curve of the three test could help in discussion.

#11) Table 1 results must be properly discussed. How the E index difference is obtained? Why is referred to only t=0s. please justify this.

#12) Line 396. Authors state in conclusion section “Future work includes the validation of this E-index method,”. Please explain this deeply to avoid misunderstanding and be more precise in this point.

Minor changes

#1) Line 17. Please add a blank space before parenthesis

#2) line 35. Please add a blank space before a bracket [. Please do the same in line 99

#3) Line 85. Please delete the dot after al. Do the same hereafter

#4) Please use italics for Latin expressions such as in vitro in vivo et al, etc.

#5) Line 160. Please use italics for variables such as t=0s

#6) line 189. Please delete the extra dot after al.

#7) Lines 200-202. Please delete these lines of the journal template

#8) Line 242. Please use italics for subheadings according to the journal template

#9) Reference section. Author must revise the reference section carefully; some errors can be found such as the following ones:

Refs. 10, 23 and 25, please do not use the abbreviated journal name, please use the complete journal name instead

Refs 4, 9, 14, 13, 16, 27, 32, please use capital letters for the journal name.

Refs 5, 12 and 21 and 34, pages are missed or include the paper number instead

Refs.  29, please modify properly the final page number

Author Response

We wish to thank the reviewers for their comments on our manuscript “Monitoring Osseointegration Process using Vibration Analysis”. Thank you for the thorough and critical review, the thoughtful comments, and the constructive suggestions, which helped to improve this manuscript. We have studied these comments carefully and have made major changes to the manuscript. We hope that these changes will meet with your approval. Please see the attachment. Thank you.

Reviewer 2 Report

The novelty of the submission and relevance of the contributions are under major question. The submitted manuscript and the published papers on Monitoring Osseointegration Process using Vibration Analysis are very similar. The present manuscript does not include any novel results or discussion and does not deserve a new paper.In conclusion, the innovation of the manuscript is very scarce.

Reviewer 3 Report

This paper investigated the capability of a vibration related index on detecting the dgreee of simulated osseointegration process. The cross spectrum and colormap of the normalized magnitude demonstrated significant changes along with the cure time, evidencing the application of these plots could improve the accuracy of the currently available diagnostic techniques. E-index exhibited a trend with a noticeable increase of averaged 53% against the cure time.

The manuscript is well organized. The work is interesting and the results are clear, which I think is acceptable for this journal.

Author Response

We wish to thank the reviewers for their comments on our manuscript “Monitoring Osseointegration Process using Vibration Analysis”. Thank you for the thorough and critical review, the thoughtful comments, and the constructive suggestions, which helped to improve this manuscript.

Round 2

Reviewer 2 Report

Accept as it is.